# Alterations in Growth Habit to Channel End-of-Season Perennial Reserves towards Increased Yield and Reduced Regrowth after Defoliation in Upland Cotton (*Gossypium hirsutum* L.)

**DOI:** 10.3390/ijms241814174

**Published:** 2023-09-16

**Authors:** Salman Naveed, Nitant Gandhi, Grant Billings, Zachary Jones, B. Todd Campbell, Michael Jones, Sachin Rustgi

**Affiliations:** 1Department of Plant and Environmental Sciences, Clemson University Pee Dee Research and Education Center, Florence, SC 29506, USA; snaveed@clemson.edu (S.N.); majones@clemson.edu (M.J.); 2Department of Crop & Soil Sciences, North Carolina State University, Raleigh, NC 27695, USA; 3USDA-ARS Coastal Plains Soil, Water, and Plant Research Center, Florence, SC 29501, USA; todd.campbell@usda.edu

**Keywords:** cotton, determinate/indeterminate growth, floral induction, meristem identity genes, annual/perennial growth habit, expression QTL (eQTL)

## Abstract

Cotton (*Gossypium* spp.) is the primary source of natural textile fiber in the U.S. and a major crop in the Southeastern U.S. Despite constant efforts to increase the cotton fiber yield, the yield gain has stagnated. Therefore, we undertook a novel approach to improve the cotton fiber yield by altering its growth habit from perennial to annual. In this effort, we identified genotypes with high-expression alleles of five floral induction and meristem identity genes (*FT*, *SOC1*, *FUL*, *LFY*, and *AP1*) from an Upland cotton mini-core collection and crossed them in various combinations to develop cotton lines with annual growth habit, optimal flowering time, and enhanced productivity. To facilitate the characterization of genotypes with the desired combinations of stacked alleles, we identified molecular markers associated with the gene expression traits via genome-wide association analysis using a 63 K SNP Array. Over 14,500 SNPs showed polymorphism and were used for association analysis. A total of 396 markers showed associations with expression traits. Of these 396 markers, 159 were mapped to genes, 50 to untranslated regions, and 187 to random genomic regions. Biased genomic distribution of associated markers was observed where more trait-associated markers mapped to the cotton D sub-genome. Many quantitative trait loci coincided at specific genomic regions. This observation has implications as these traits could be bred together. The analysis also allowed the identification of candidate regulators of the expression patterns of these floral induction and meristem identity genes whose functions will be validated.

## 1. Introduction

Cotton (*Gossypium* spp.) is an important industrial crop produced worldwide to meet the demand for natural fibers. Since the global population is expected to reach nine billion by 2050 [1], a concomitant increase in the demand for food and fiber is anticipated. Cotton is grown commercially in tropical and sub-tropical regions of more than 70 countries, and it is the only crop where the fabric is made by ginning fiber at an industrial scale [2]. According to the National Cotton Council (www.cotton.org; accessed on 11 May 2022), about 75% of cotton fiber is used for apparel products, 18% for home furnishings, and 7% for industrial products. Cotton yield has increased significantly in the past century but has stagnated over the last decade. In addition, the extreme weather conditions and climatic-change-driven alterations in the insect pest and pathogen populations have further impacted cotton yield [3]. Therefore, novel strategies are needed to increase cotton productivity to meet future global demand.

Cotton exists as a diploid and a polyploid and was domesticated in the old and new world independently from perennial shrubs or trees [4]. The new-world domesticated cotton species include “Upland cotton” (*G. hirsutum*) and “Egyptian or Sea Island” cotton (*G. barbadense*) [5], and the old-world cotton consists of “tree cotton” (*G. arboreum*) and “levant cotton” (*G. herbaceum*) [6]. Among 52 *Gossypium* spp., including seven tetraploids with the “AD” genome and 45 diploids with “A–G” and “K” genomes [7,8], only four species mentioned above were domesticated from photoperiod-sensitive perennial species into day-neutral types. However, cotton production today primarily relies on allotetraploid cotton, *G. hirsutum* (2*n* = 4x = 52), accounting for over 90% of the world production, and the remainder comes from *G. barbadense*, and to a much lesser extent from two diploids *G. arboreum* and *G. herbaceum*. Allotetraploid cotton, *G. hirsutum* with “AD” genomes (with a genome size of 2.4 Gb comprised of 26 chromosomes), evolved from crossing and polyploidization of two diploid species with “A” and “D” sub-genomes approximately 1–2 million years ago [7]. The 26 chromosomes in the Upland cotton genome are numbered according to genome ancestry, where chromosomes 1–13 belong to the “A” sub-genome and chromosomes 14–26 to the “D” sub-genome [7,9].

Phenotypic variation in cotton is polygenically controlled for most of the agronomic traits [10]. The genetic and molecular bases of these variations are crucial to determining the relationship between genotype and phenotype. Plant architecture is an important characteristic in determining crop productivity. Different traits like plant size and shape, positioning of leaves and branches, time of appearance and distribution of reproductive structures affect crop management and production in cotton [11], and these traits can be modified to improve crop yield and quality [12]. Plant architecture is also influenced by its growth habit. Crop plants typically have two types of growth habits, namely determinate and indeterminate. A determinate growth habit is desirable in crop plants with shorter growing seasons, where prolonged vegetative growth leads to yield losses, e.g., in wheat, tomato, cotton, soybean, potatoes, and strawberries. In addition, it allows mechanical harvesting. Therefore, cultivars with a determinate growth habit have been preferred in several crop plants [13,14,15]. It is reported that crop domestication has favored more determinate plant architecture (i.e., compact growth habit) with a synchronous flowering time that supports higher yields than indeterminate growth habit. Thus, a detailed understanding of the floral transition of the vegetative meristem, i.e., the transition from indeterminate to determinate growth, could be highly beneficial for improving crop productivity.

With the advancement in molecular genetics and quantitative trait loci (QTL) mapping approaches, major effect genes have been identified that are involved in molecular mechanisms contributing to diverse plant morphological traits [16,17,18,19,20]. Genes determining plant architecture, such as *FLOWERING LOCUS T* (*FT*), *FRUITFUL* (*FUL*), *APETALA1* (*AP1*), *CAULIFLOWER* (*CAL*), and *WUSCHEL* (*WUS*), were studied in detail, and their molecular functions deciphered [21]. Additionally, different mapping techniques are currently being used to fine-map QTLs associated with plant architecture traits in different crop plants, including cotton [22,23,24,25].

The switch from the vegetative to the reproductive phase in plants is a fundamental process that plays a vital role in plant adaptation and crop productivity [26,27,28,29,30,31,32,33,34]. Phenological traits such as the timing of floral transition are of high significance in wild and domesticated plants, and identification of the genes controlling this trait has been an active area of research for the last couple of decades. In *Arabidopsis thaliana*, about 90 genes comprise the network of flowering genes, most of which exist in small multigene families (e.g., phytochrome and *AGAMOUS*-like genes). *Flowering locus T* that encodes a transmissible protein known as “florigen” is one of the most widely studied genes with homologs identified in rice (*Oryza sativa*) [35], wheat (*Triticum aestivum*) [36], barley (*Hordeum vulgare*) [37], sunflower (*Helianthus annuus*) [38], pea (*Pisum sativum*) [39], and chrysanthemum (*Leucanthemum maximum*) [40]. The genomes of three diploid cotton and two tetraploid cotton were sequenced recently [41], providing the framework for bioinformatic identification of orthologs of the flowering time genes in cotton. It is known that the wild accessions and landraces of the cotton possess a significant amount of genetic diversity and provide excellent resources for breeding resistance to pests and pathogens, early maturity, high yield, and determinate growth habit or compact plant architecture in Upland cotton [42,43].

Cotton is primarily a self-pollinated perennial crop with an indeterminate growth habit. The perennial nature and indeterminate growth habit in cotton pose specific challenges in cotton production, the most significant being the regrowth after defoliant application. As cotton bolls mature and dehiscence, they no longer draw nutrients from the photosynthetic source tissues (i.e., leaves, stems, and bracts), which causes photosynthates to accumulate in the plant. This accumulation of photosynthates often induces a second round of vegetative growth, commonly known as “regrowth” (https://www.cropscience.bayer.us/articles/dad/cotton-regrowth-management; accessed on 4 September 2023). Regrowth after defoliation causes serious problems for cotton growers by not only reducing fiber quality from increased lint staining, increased trash, and reduced harvest efficiency, but also constituting food sources for insect pests and a host to various pathogens. In addition, massive use of defoliants [tribufos (a cell wall disrupter), diuron (a photosynthesis inhibitor), thidiazuron (an auxin inhibitor), and ethephon (a plant growth regulator)] raises environmental and public health concerns. Given these challenges, cotton cultivars with an annual/determinate growth habit, optimal flowering time, and compact architecture are needed to minimize yield and economic losses. To dissect the genetics of these interrelated traits, i.e., perennial vs. annual growth habit, determinate vs. indeterminate growth, and regrowth after defoliation, we evaluated the Upland cotton mini-core collection of 44 genotypes [44] for the expression pattern of five floral induction and meristem identity genes, *FT*, *SOC1*, *LFY*, *API*, and *FUL*.

In *Arabidopsis*, the two flowering time genes, *SUPPRESSOR OF OVEREXPRESSION OF CONSTANS 1* (*SOC1*) and *FUL*, were identified to control the annual growth habit [45]. Also, *FT* that encodes a mobile signal protein, “florigen”, was demonstrated to regulate the balance between indeterminate and the determinate growth habit and hence the plant architecture besides floral transition. *FT* regulates plant architecture through the meristem identity genes *LEAFY* (*LFY*) and *AP1* that initiate flower development [46,47,48].

This study aims to create resources for the development of high-yielding cotton cultivars exhibiting a more determinate growth habit and reduced regrowth after defoliation application. Key objectives of this study are (1) identification of high-expression alleles of the five floral induction and meristem identity genes in Upland cotton genotypes; (2) mapping of the expression QTLs (eQTLs) responsible for the expression level differences observed for these five genes among Upland cotton genotypes using a genome-wide association study; (3) analyzing the diversity of the Upland cotton genotypes used in this study; and (4) identification of candidate genes responsible for the eQTLs. We explore these objectives and their outcomes in the following sections.

## 2. Results

### 2.1. Gene Expression Analysis for the Floral Induction and Meristem Identity Genes

To identify and stack the high-expression alleles of the five floral induction and meristem identity genes in a single genetic background, we chose to screen a mini-core collection of 44 Upland cotton genotypes. From 2017 to 2019, we screened this mini-core collection for the high-expression alleles of *FT*, *SOC1*, *LFY*, *AP1*, and *FUL* cotton genes at three developmental stages using qRT-PCR. Collectively, we recorded data for 45 gene expression traits, i.e., five genes and three developmental stages, for three consecutive years (5 × 3 × 3). All genes were recorded similarly, and the gene expression data were normalized to *ACT4-2* control. We created an arbitrary expression matrix to analyze the gene expression data and categorize genotypes (see Section 4 for details). Eight genotypes received a range of 19–34 points, which is more than the remaining studied genotypes, and hence these genotypes were selected to make genetic crosses (Table 1).

Further, we studied the trait distribution of the gene expression traits (e-traits). The gene expression data showed a normal distribution for most e-traits. An example of the distribution patterns observed for Stage 1, 2019 e-traits is provided in Appendix A. Overall, Stage 1 samples showed a relatively low expression of most of the genes than Stage 2 and Stage 3 samples. Among five genes, *AP1* consistently showed a higher expression level in all the developmental stages except S1 in 2017 and 2019, followed by *FT*, whereas *SOC1* consistently showed a lower expression in most developmental stage year combinations.

### 2.2. Genome-Wide Association Mapping of the Gene Expression Traits

DNA extracted from 44 diverse Upland cotton genotypes was genotyped for 45,104 intraspecific SNP markers. After analysis, a call rate in the range of 91.9–99.7% was observed in different genotypes, which was above the threshold call rate of 90% set for further analysis. The data were analyzed for polymorphism paying attention to 38,822 polymorphic SNPs identified earlier [49]. Over 14,520 SNPs (about six markers per Mb) showed polymorphism and were used further to study associations with 45 e-traits (i.e., the expression pattern of the cotton *FT*, *SOC1*, *LFY*, *AP1*, and *FUL* genes recorded over three developmental stages, S1, S2, and S3 in three years, 2017–2019). Three hundred ninety-six markers distributed over all chromosomes showed association with e-traits (Figure 1). Over 64% of associated markers were cis-eQTLs (±1 Mb of the genes of interest). Many eQTLs coincided at specific genomic locations on chromosomes A01, A12, D01, and D05. An eQTL (i34012Gh) from the non-genic region was consistently detected over the years, and three eQTLs (i18496Gh, i02927Gh, and i14752Gh) were consistently detected at different developmental stages in a year.

One hundred and fifty-nine of e-trait-associated markers were identified to have their origins in genes; 24 and 26 markers, respectively, belonged to 5′- and 3′-untranslated regions (UTRs), and 187 markers were from non-coding random genomic regions (Appendix A). Out of these 396 e-trait associated markers, 188 markers mapped to the A subgenome, and 208 markers mapped to the D subgenome. A maximum of 31 and 26 e-trait-associated markers mapped to chromosomes D05 and D12, respectively, and a minimum of 1 marker mapped to chromosomes A06. An SNP marker, i30101Gh, of the non-genic region showed association with the e-traits of all five floral-induction and meristem identity genes, 14 markers showed association with four e-traits, 31 markers with three e-traits, 82 markers with two e-traits, and the remaining markers showed association with a single e-trait each. Biased distribution of the e-trait-associated markers to different A and D subgenome chromosomes was observed (Appendix A). An expected number of associated markers was calculated per chromosome to determine the biased distribution of associated markers on different chromosomes. For this purpose, a random and even distribution of markers across the genome was assumed. When the expected number of markers was plotted with the observed number of associated markers on each chromosome, a biased distribution of associated markers on different cotton chromosomes became apparent (see Appendix A).

### 2.3. Linkage Disequilibrium (LD) and LD Decay

Linkage disequilibrium (LD) was calculated for 396 markers that showed association with e-traits. Overall, 370 markers were found in LD, where 175 markers belonged to the A subgenome and 195 to the D subgenome (Appendix A). The lengths of LD blocks varied among chromosomes. Within each chromosome, the LD blocks differed in size depending on the chromosome regions; for example, in the case of chromosomes A08 and D11, large LD blocks were observed in centromeric and telomeric regions, respectively. From the A subgenome, chromosomes A06 and A13 had the lowest and highest numbers of e-trait-associated markers, and most markers were in LD and originated from the telomeric region of the chromosome (Appendix A).

### 2.4. Population Structure and Diversity Analysis

We performed population structure analysis to estimate the genetic diversity among 44 Upland cotton genotypes of the mini-core collection and used the marker data set to investigate the relationship of the mini-core collection with the core collection of 382 accessions [44]. The analysis revealed seven population groups indicating that the mini-core collection had genes from seven different populations, reflecting the level of diversity among 44 Upland cotton genotypes. Out of 44 accessions, 18 had admixtures from two or more population groups, and 2 had admixtures from four different populations (Figure 2A).

Additionally, we constructed the dendrogram to find out the interrelationships of 44 Upland cotton genotypes, paying specific attention to the 8 genotypes selected based on the expression analysis of five floral induction and meristem identity genes, *FT*, *SOC1*, *LFY*, *FUL*, and *AP1* to make genetic crosses (see above). The dendrogram clustered them into three distinct groups (Figure 2B). The overall trend of their interrelatedness was similar to the pattern observed in the population structure analysis. For example, the genotypes HOPI MOENCOPI and GSA-74 in the structure analysis showed maximum diversity and clustered at two extremes in the phylogenetic analysis. Likewise, COKER-201 was clustered closely with HOPI MOENCOPI in the phylogenetic analysis (Figure 2B) and the population structure analysis (Figure 2A). The genotypes CABD3CABCH-1-89, CAHUGLBBCS-1-88, SPNXCHGLBH-1-94, and TAMCOT SP-23 grouped in the same cluster exhibiting close relationship in the population structure analysis and pairwise identity matrix of the parental genotypes. These results shed light on the genetic relationships of the Upland cotton mini-core collection genotypes, which has wide implications in making genetic crosses and marker-assisted selection.

Further, to delve into the genetic representation of a larger collection of cotton genotypes ever bred and cultivated in the Southeastern US, we performed diversity analysis using the identity-by-state (IBS) method. The genotypes with orange nodes in Figure 2C represent the mini-core collection, the purple nodes represent the genotypes of the Pee Dee germplasm collection, whereas the black circles represent the other improved germplasm under cultivation in the continental US. This analysis suggested that 44 diverse Upland cotton genotypes of the mini-core collection represent ~92% diversity in the core collection of genotypes cultivated in the Southeastern United States in the past century. The level of representation is satisfactory and endorses the choice of this material for gene expression analysis performed in this study. However, the level of diversity and small population size were expected to pose a challenge in finding associations with the e-traits, which should not cause a terrible bottleneck due to the small genetic base of the US cotton germplasm and inherently low level of genetic diversity.

Moreover, principal component analysis was performed to elaborate further on the genetic relationships of the 44 Upland cotton mini-core collection genotypes. For this purpose, the first two principal components were plotted; it grouped the 44 Upland cotton genotypes into six main clusters. Out of 44 genotypes, 29 belonged to Population one, 5 belonged to Population 2, 3 belonged to Population 3, 1 belonged to Population 4, and 4 and 2 genotypes, respectively, to Populations 5 and 6 (Appendix A). Largely, the clustering of the genotypes was the same as observed in the population structure analysis. For instance, CABD3CABCH-1-89, SPNXCHGLBH-1-94, and CAHUGLBBCS-1-88 clustered together in a single population. Further, it was noticed that the genotypes selected for genetic crossing in this study had positive ordination, which suggests their suitability for heterosis breeding. In sum, out of eight selected genotypes, seven belonged to PC1 and exhibited higher positive scores.

### 2.5. Pedigree Analysis

We performed pedigree analysis to interpret the diversity analysis results and understand the admixtures between populations. As some genotypes of the mini-core collection are very old, collecting information about their parentage became a challenge. We were able to find the pedigrees of 41 of the 44 genotypes, and only incomplete information could be obtained for ALLEN 33, BJAGL NECT, and M.U.8B UA 7-44; hence, these genotypes were treated as introductions. A pedigree diagram was developed using the pedigree information for the genotypes of the mini-core collection (Appendix A). The genotypes with yellow nodes in the pedigree diagram represent 44 Upland cotton genotypes used in the present study. The pedigree analysis exhibited the relationship among genotypes clustered in phylogenetic analysis and/or structure analysis, hence explaining the cause of the observed clustering of genotypes in the phylogenetic analysis. For example, ARKOT-8102 (ST-825/Miscot-T8-27) in the structure analysis was revealed to have admixtures from three different populations. The pedigree of this genotype also showed it to carry the genetics of dozens of parental genotypes. Similarly, GSA-74 (a selection from HYC-MDR-2) showed to carry the genetics of several parental genotypes. Using plants with complete pedigree information helps identify the source of alleles and trace the descent of a trait [50,51].

### 2.6. Gene Family Analysis

We performed gene family analysis for the five floral induction and meristem identity genes, *FT*, *SOC1*, *LFY*, *FUL*, and *AP1*, to understand the relationships of different gene family members and to reflect on their transcriptional regulation. We identified two copies of *FT*, thirteen of *SOC1*, two of *LFY*, and six of *FUL* and *AP1* genes in the *G. hirsutum* genome using sequence similarity searches and used them for the gene family analysis. Hence, in this study, we are reporting two copies of the *FT* gene in the cotton genome, *Gohir.A08G227700* (l,125,104,463 bp to 125,101,112 bp) and *Gohir.D08LOC107909115* (67,103,565 bp to 67,100,103 bp) on chromosomes A08 and D08, respectively; likewise, two copies for the *LFY* gene, *Gohir.A07G050800* (5,562,304 bp to 5,557,891 bp) and *Gohir.D07G050800* (6,018,761 bp to 6,014,246 bp) on chromosomes A07 and D07, respectively, and more than two copies for *SOC1*, *FUL*, and *AP1* genes. We pulled out SOC1, FUL, and AP1 protein sequences from the public domain and used them to develop the cladograms using the default settings on the Emboss software v. 6.6.0 (https://www.ebi.ac.uk/Tools/msa/clustalo/; accessed on 29 November 2022). *SOC1* gene family members are divided into two major clusters, where common evolutionary origins were observed for *Gohir.A13G058601* and *Gohir.D12G081801* genes, whereas *Gohir.LOC107957281* and *Gohir.A12G102700* were more distantly related in the phylogenetic tree (Appendix A).

### 2.7. Promoter and miRNA Analysis

Promoter sequences (1 kb region upstream to the transcription start site) of the five genes of interest (*FT*, *SOC1*, *LFY*, *FUL*, and *AP1*) were retrieved from Phytozome and analyzed using default parameters in the PlantPan 3.0 software. Different transcription factor binding sites were predicted in the sequences, and some TFs had target sites in all genes of interest. We identified binding sites for up to 34 TFs belonging to as many as 12 TF families to co-occur in the 1 kb region upstream to the genes of interest. For example, the GATA transcription factor family that has target sites in the promoters of all five genes of interest is shown in Appendix A.

### 2.8. Determination of the Molecular Functions of the Trans-eQTLs and the Putative Causes of Epistatic Interactions between the eQTLs and the Genes of Interest

Out of a total of 396 e-trait-associated SNP markers, we studied 209 markers belonging to coding sequences (CDS), mRNA, and 5′/3′-UTRs to find out their epistatic interactions with genes of interest. Out of 209 e-trait-associated SNP markers, 77 were possibly involved in post-transcriptional regulation of gene expression, 44 in post-translational regulation, and 36 encode enzymes involved in epigenetic regulation, whereas 52 were annotated as uncharacterized proteins (Appendix A). In some cases, multiple e-trait-associated SNPs (eQTLs) map to a single transcription factor; for example, SNP markers i12578Gh, i12579Gh, i20152Gh, and i12580Gh in chromosome D04 identified as eQTLs for the cotton *FUL* gene (*Gohir.D04G047400*; location: 7.84782 Mb to 7.85123 Mb) map to a KH-domain containing protein At4g18375-like transcription factor. Interestingly, some eQTLs, i.e., the SNP markers i18496Gh and i14751Gh in chromosome A01 (location 117.05121 Mb to 117.05165 Mb), showed a pleiotropic effect by controlling the expression of multiple genes i.e., *FT*, *LFY*, and *AP1*. These SNPs mapped to a gene encoding a transcription factor of the zinc-regulated protein 8-like family.

Further, we attempted to establish the interaction network between the eQTLs based on the gene annotations and binding sites of specific transcription factors and target sites of micro RNAs in the promoter regions or the gene bodies of *FT*, *SOC1*, *LFY*, *FUL*, and *AP1* cotton genes. This analysis identified 8 SNP markers (Table 2) mapped to transcription factors with sites in promoter regions of the corresponding genes.

We also analyzed the gene bodies and promoter regions of the 29 genes related to *FT*, *LFY*, *AP1*, *SOC1*, and *FUL* for miRNAs and miRNA target sites. Likewise, gene bodies of 209 e-trait-associated SNPs were searched for pre-miRNAs (Appendix A). The pre-miRNAs and miRNA target sites in the gene bodies of most of the gene family members were absent except *Gohir.D07G071700*, *Gohir.D07G084400*, *Gohir.A07G067100*, *Gohir.A07G079100* (*FUL*), and *Gohir.D11G008200* (*SOC1*). Likewise, target sites for miRNA were absent from the promoter regions of most gene family members except *Gohir.A08G097662* (gra-miR8709b) and *Gohir.D03G097900* (gra-miR8716). Furthermore, out of a total of 209 eQTLs (genes with e-trait-associated SNPs), gene bodies of 40 eQTLs showed similarity to pre-miRNAs, where pre-miRNA MIR8739 showed maximum similarity in *Gohir.D06G215500*, an eQTL for *AP1* e-trait (Appendix A). Similar to the interaction network of TF and TF-binding sites in the promoter region of the genes of interest, we studied the correspondence between eQTLs mapping to miRNAs and miRNA target sites in the promoter regions of five target genes and their gene family members. For instance, a miRNA target site exists in the promoter region of the cotton *AP1* gene (*Gohir.A08G097662*), and the eQTL (SNP marker i08618Gh in *Gohir.A12G250300*) for this gene maps to a corresponding pre-miRNA sequence in chromosome A12. Likewise, a miRNA target site exists in the gene body of the cotton *FUL* gene (*Gohir.D07G071700*), and the eQTL (SNP markers i20989Gh and i20988Gh in *Gohir.D11G323100*) for this gene maps to a corresponding pre-miRNA sequence in chromosome D11 (Appendix A).

### 2.9. CpG Island Prediction in the Members of FT, SOC1, LFY, FUL, and AP1 Gene Families

We screened the gene bodies and the promoter regions of the *FT*, *SOC1*, *LFY*, *FUL*, and *AP1* gene family members for the CpG islands (CGIs) using the CpGPlot v. 6.6.0 (https://www.ebi.ac.uk/Tools/seqstats/emboss_cpgplot/; accessed on 21 December 2022) (Appendix A). The promoter regions (sequence 1kb upstream of the start codon) of none of the genes were predicted to have a CGI, whereas the gene bodies of *Gohir.D07G050800* and *Gohir.A07G046500* were predicted to have CGIs at two different positions in the sequence. The length of these CGIs was 338 and 300 bp for *Gohir.D07G050800* and 336 and 299 bp for *Gohir.A07G046500* (Appendix A).

### 2.10. Expression Analysis of the Members of FT, SOC1, LFY, FUL, and AP1 Gene Families

The whole gene sequences of *FT*, *SOC1*, *LFY*, *FUL*, and *AP1* cotton gene family members were compared with the expressed sequence tags (ESTs) to determine the expression patterns of these genes in different organs. An arbitrary tissue-based matrix was developed based on the number of blast hits against a gene sequence to highlight whether a gene expresses in a specific tissue and to what extent. The hits to a gene sequence were divided based on the source tissue/organ, and the number of hits per tissue/organ was counted and provided with a unique color in the gene expression heatmap (Appendix A). The columns in the heat map represent tissues of expression, and the rows represent the genes. Based on this in silico expression analysis, most of the genes exhibited a low expression level (one to six ESTs) except for the *SOC1* gene (*Gohir.D02G198000*), which exhibited >13 hits. Further, the *SOC1* gene family, which was the largest with 13 members, also had members like *Gohir.D02G198000*, which express in many tissues (leaf, anther, root, cotton ovules, etc.) other than *Gh_LOC107957281* that express exclusively in shoot, apical meristem, buds, and flower (SAMBF). Interestingly, one *AP1* gene (*Gohir. D04G190200*) and none of the *LFY* genes showed similarity with EST sequences.

## 3. Discussion

Finding eQTLs/genes contributing favorably to the cotton growth habit (annual/perennial and indeterminate/determinate) that reduce regrowth after defoliation is challenging due to its complex genome and negative correlation between early maturity and lint yield, as well as early maturity and fiber quality [52,53,54]. Therefore, it is difficult to simultaneously improve these traits using conventional breeding methods alone, and identifying associated DNA markers is vital for simultaneously improving such traits via marker-assisted selection (MAS). Association mapping is one such method that allows the identification of the favorable maker alleles that facilitate breeding for complex traits. Favorable alleles for yield and its components and the fiber quality traits were identified earlier using association analysis in Upland cotton [55,56]. In this study, we performed genome-wide association analysis for the 45 e-traits of five floral induction and meristem identity genes (*FT*, *SOC1*, *LFY*, *FUL*, and *AP1*) across the Upland cotton genome. We genotyped 44 Upland cotton genotypes of the mini-core collection that represent ~92% diversity in a much larger collection of the Upland cotton genotypes ever bred in the US. One concern this population raises is its small size, which is expected to make it difficult to find associations. Another challenge is the low heritability and influence of the environment on gene expression traits. The first criticism is not so concerning; as evident from the pedigree analysis, the Upland cotton has a narrow genetic base, and most of the cultivars shared some heritage during their development, which will help identify marker-trait associations. The low heritability of the e-traits was dealt with by performing the experiment for three consecutive years in the same field from the same source of seeds with minimal to no change in the management practices including sample collection and analysis procedures. Identification of consistent eQTLs, which at times were not the same markers but a marker in LD with the markers initially showed association with an e-trait, indicated the successful use of a manageable population for such a study. Expression QTL analysis was successfully applied in many crop plants earlier [57,58,59,60,61,62,63,64,65,66], but mostly using bi-parental populations or largely in diploid organisms. Hence, this study presents precedence to perform it in a polyploid crop and for the gene(s) of interest, not the whole transcriptome making it more affordable. Each sample was genotyped for 45,104 intraspecific SNPs, and some of these markers produce redundant information, as in cotton, the LD extends from 100 kb (chromosome 19) to 5750 kb (chromosome 25) [67], which is somewhat similar to the level of LD decay observed in the present study. With this level of LD, it appears the number of markers used in this study is more than sufficient.

Many genes controlling early maturity and fiber quality in cotton have been reported previously. Further, chromosome D03 has been reported earlier to be rich in QTLs for early maturity traits [68,69,70,71]. Interestingly, we also found an SNP marker, i03389Gh, in chromosome D03, which maps to a gene encoding FRIGIDA-like protein transcription factor that regulates flower development in cotton. These findings suggested that chromosome D03 contributes to the early maturity and growth habit-related traits in cotton. Moreover, through a detailed examination of molecular functions assigned to the eQTLs (e-trait associated markers mapping to genes) and the promoter analysis of the genes of interest, we identified six candidate genes possibly involved in cotton floral induction and meristem identity, hence growth habit [72]. One out of six candidate genes, *Gohir.A01G208700* on chromosome A01 was identified to regulate three e-traits (*AP1*, *FT*, and *LFY*), and two genes (*Gohir.A13G0504000* on chromosome A13 and *Gohir.D12G153600* on chromosome D12) to regulate two e-traits each, *FT* and *LFY*, and *FUL* and *SOC1*, respectively.

In the present study, we noticed that the expression level of floral induction and meristem identity genes was the lowest at Developmental stage 1. It was also apparent from the above discussion that *FT* expresses highly during the reproductive stage or during the vegetative to floral transition in cotton plants. Complementing these ideas, the expression of these genes was the highest at the second developmental stage and relatively lowered at the third developmental stage. Further, it appears from the review of the literature that the floral induction and meristem identity genes exhibit conserved developmental stage-specific expression patterns in different crop plants [73,74,75].

We also evaluated the gene and promoter sequences for CGIs during the present study. CpG islands in the promoter or gene body are known to regulate gene expression through transcriptional silencing of the corresponding gene [76]. It was reported earlier that promoters usually lack DNA methylation; in contrast, genic bodies show high degrees of DNA methylation because of CGIs [77,78]. Interestingly, in our analysis, none of the promoter sequences predicted to have a CGI (Appendix A). On the other hand, bodies of 37 (13.7%) genes (with e-trait-associated SNP markers) were predicted with CGIs. These observations might reflect on the methylation and transcriptional status of the gene with a likely impact on cotton growth and development.

Natural epialleles have been studied in different crop plants (e.g., rice and tomato), including cotton, and are shown to affect various phenotypic traits [79,80]. In tomato, the epimutant *cnr* of the *CNR* gene is transcriptionally repressed by DNA hypermethylation in the promoter region, resulting in a colorless, non-ripening fruit phenotype [79]. In the case of rice, the epiallele mutant *rav6*, the DNA hypomethylation of the promoter causes ectopic expression of RAV6 (RELATED TO ABSCISIC ACID INSENSITIVE 3/VIVIPAROUS1 (VP1) RAV FAMILY GENE 6) that affects the leaf angle and seed size in rice, alters the leaf angle by modulation of the brassinosteroid homeostasis [81]. Similarly, in allotetraploid cotton, the *CONSTANS-LIKE 2D* (*COL2D*) gene is hypermethylated, and its epiallele in wild cotton is hypomethylated, which promotes flowering [82].

In cotton, the rate of vegetative growth, i.e., adding more mainstem nodes with subtending leaves, flowers, or fruiting branches, slows down as boll development begins. Later in the developmental process, a time approaches when vegetative growth completely ceases. This specific developmental time point in cotton is known as “cutout”. From this transition point in development, the cotton crop becomes more source limited, i.e., limited by assimilates (carbohydrates), in contrast to the developmental stage when there is less vegetative growth and a surplus of resources (https://www.cotton.org/tech/ace/upload/Integrated-Crop-Management.pdf; accessed on 4 September 2023).

An optimally yielding cotton variety generally produces 25 mainstem nodes, and 18 of them develop into fruiting branches (https://extension.uga.edu/publications/detail.html?number=C1244&title=cotton-growth-monitoring-and-pgr-management; accessed on 4 September 2023). After cutout in cotton, generally, young fruits exhibit shedding due to competition for resources with maturing fruits. Hence, in cotton, more than 90% of fruits from the first fruiting branches are retained compared to less than 20% from the upper branches. On the other hand, the climatic conditions, heat stress, drought, and erratic rain patterns cause cotton plants to shed many fruits, often reversing to vegetative growth (regrowth). For instance, extensive regrowth was seen in years that support lower yields and poor boll set.

Taking note of these natural processes in cotton modulators of phytohormones, such as Mepiquat chloride, a gibberellin biosynthesis inhibitor was used to reduce vegetative growth, i.e., plant height and branching. Vegetative growth restriction by Mepiquat induces the plant to direct more carbohydrates into reproductive organs, hence aiding in approaching an early cutout to support developing bolls [83].

A similar balance between vegetative and reproductive growth, or, in other words, the potential to regrowth from unutilized resources in the case of early cutout cotton or plants with ceased vegetative growth, can be achieved by ectopically expressing the *FT* gene. Indeed, McGarry and Ayre [84] transiently expressed *GhSFT* in photoperiod-sensitive “Texas 701” and day-neutral “DeltaPine 61” using viral vectors. In both genetic backgrounds, cotton shoot architecture turned more determinate compared to control, where sympodial branches initiated from earlier nodes of the main stem and terminated prematurely with clusters of flowers instead of initiating the next sympodial unit [84]. Similarly, virus-induced gene silencing (VIGS) of *GhSP* in wild and cultivated cotton resulted in strongly determinate plants, where no monopodial vegetative or sympodial fruiting branches were produced as all vegetative and monopodial shoot apical meristem of the main axis and axillary meristems terminated into flowers [85].

These phenotypes are way more extreme for developing cultivars, partitioning end-of-season reserves moderately between vegetative and reproductive growth to maximize yield while preventing regrowth. Instead, the quantitative genetic approach adopted in the present work will allow fine-tuning the balance between vegetative and reproductive growth by identifying and selecting the regulators of the five major floral induction and meristem identity genes, *FT*, *SOC1*, *AP1*, *FUL*, and *LFY*. Among these genes, *FT* expresses in leaves, and the protein moves through the phloem and reaches the meristem. In meristems, *FT* interacts with transcription factor FD and 14-3-3 proteins to form a florigen activation complex to stimulate expression of *SOC1* and *FUL* and meristem identity genes *AP1* and *LFY* to determine the fate of meristem to either continue vegetative growth or develop into a flower [85]. Expression QTLs identified in this work will allow modulating their expression, and the identified associated markers will facilitate staking their effects to variable degrees to tailor genotypes suitable for a production region with a long or a short growing season.

## 4. Materials and Methods

### 4.1. Plant Materials and Sample Collection

Seeds of 44 diverse Upland cotton genotypes of the mini-core collection were available at the Pee Dee Research and Education Center (PDREC), Florence, SC. For names and the field ID of each genotype, see Appendix A. Plants were cultivated at the same research field (34°18′39″ N 79°44′40″ W) in PDREC for three consecutive years (2017 to 2019) to collect plant tissue for RNA extractions. Each accession was grown in two 40 feet rows where individual plants were space planted with a 1-foot plant-to-plant and row-to-row distance. As stated, the experiment was performed at the same field location for three years to avoid, as much as possible, edaphic, management, and ecological variabilities (insect pest and pathogen pressure), as the gene expression traits were shown earlier to be highly influenced by these factors [86]. Three plants, specifically somewhat from the center of the two rows showing the characteristics of specific genotypes, were tagged and used for taking plant samples to serve as three biological replicates in the experiment. Plant tissues were collected at three different developmental stages at 10, 30, and 45 days after sowing (DAS), designated as Stage 1 (S1), Stage 2 (S2), and Stage 3 (S3), respectively, from the young cotyledonary leaves (S1) and young squares (S2, S3), from the field between 7 to 10 A.M. The tissues were collected on liquid nitrogen in the field and immediately stored in a −80 °C freezer until used for RNA extraction. Samples were also collected from three weeks old true leaves for DNA extraction, followed by genotyping using cotton 63 K SNP Array.

### 4.2. Gene Expression Analysis

Total RNA was extracted by two different methods: (i) manually with the Spectrum™ Plant Total RNA Kit (Sigma-Aldrich, Inc., St. Louis, MO, USA), and (ii) mechanically with the KingFisher Flex System using the MagMAX Plant RNA Isolation Kit (Thermo Fisher Scientific, Waltham, MA, USA). The first method was applied to the samples collected in the first two years, and the second method was applied to the samples collected in the third year. Total RNA was extracted in either of the methods following the manufacturer’s instructions. After extraction, the RNA quantity and quality were determined using NanoDrop™, a UV–Vis spectrophotometer (Thermo Fisher Scientific, Waltham, MA, USA), and tested for the absence of genomic DNA contamination by PCR using the cotton *ACTIN 4-2* (*ACT4-2*)-specific primers. Subsequently, RNA was converted to cDNA for the quantitative reverse transcription PCR (qRT-PCR) analysis using the RevertAid First Strand cDNA Synthesis Kit (Thermo Fisher Scientific, Waltham, MA, USA) following the manufacturer’s instructions. The qRT-PCR analysis was performed on an iQ^TM^5 real-time PCR detection system (Bio-Rad Laboratories, Hercules, CA, USA) using SYBR^®^ Green Master mix (Bio-Rad Laboratories, Hercules, CA, USA). Gene-specific primers for the five floral induction and meristem identity genes (*FT*, *SOC1*, *FUL*, *LFY*, and *AP1*) and a housekeeping gene *ACT4-2* for normalization of the expression data were synthesized and used in qRT-PCR analysis [87]. The list of the primers used in this study is provided in Appendix A. As mentioned earlier, three biological and two technical replicates were used for the expression analysis, and the data were analyzed using the 2^–∆∆Ct^ method [88]. The expression data recorded as above was used as a phenotype to map expression QTLs (eQTLs).

Later, the gene expression data were also used to develop a gene expression matrix to grade genotypes based on their expression profiles and select genotypes with complementing high-expression alleles of the genes of interest for genetic crossing and stacking of the expression traits. For this purpose, we developed a point system where each individual received a point for an expression trait when it was found to express a gene at a developmental stage in a given year more than the population mean of that gene’s expression. All genes were recorded similarly. In this way, an individual can receive a maximum of 45 points (5 genes × 3 developmental stages × 3 years).

### 4.3. Genomic DNA Extraction and SNP Genotyping

Genomic DNA was extracted from the true leaves of three-week-old plants using the DNeasy Plant Mini Kit (Qiagen Inc., Germantown, MD, USA) following the manufacturer’s instructions from the samples collected in the 2018 growing season. DNA quantity and quality were determined using a NanoDrop Spectrophotometer (Thermo Fisher Scientific, Waltham, MA, USA) and 1% (*w*/*v*) agarose gel. DNA concentrations were adjusted to 100 ng/µL and were sent on dry ice to Texas A&M Institute for Genomic Sciences and Society (College Station, TX, USA). Later, at Texas A&M, the samples were reevaluated for quality using the PicoGreen assay [89], and the sample DNA concentrations were adjusted to 50 ng/µL. Subsequently, the calibrated DNA samples were hybridized to the SNP probes of the Cotton 63K SNP array in a custom Infinium iSelect HD Genotyping Assay (Illumina Inc., San Diego, CA, USA) following Hulse-Kemp et al. [49].

### 4.4. Diversity Analysis

Diversity analysis of the 44 Upland cotton genotypes was performed using MEGAX v. 10.1.8 [90], RStudio v. 1.2.5042 (RStudio, Inc., Boston, MA, USA), and fastSTRUCTURE [91]. A neighbor-joining method was used to construct the phylogenetic tree with 10,000 bootstraps and default parameters. The fastSTRUCTURE was run for K = 1–10 using default settings. We also considered the pedigrees of the cotton genotypes. For the pedigree analysis, we used Helium v. 1.19.09.03 and ran it with default settings [92].

### 4.5. Genome-Wide Association Study (GWAS) and eQTL Analysis

Each DNA sample was genotyped for 45,104 intraspecific SNP (single nucleotide polymorphism) markers for association analysis. The data were analyzed for alleles at 38,822 polymorphic SNPs reported in Hulse-Kemp et al. [49]. PLINK v 1.07 was used to sort SNPs based on polymorphism [93], minor allele frequency (MAF) of >2.5%, and a call rate (CR) of >90% to develop a set of SNPs for association analysis. Subsequently, RStudio v. 1.2.5042 [94] was used for the genetic mapping using the SNP set mentioned above. The haplotypes were identified using Haploview v. 4.2 with default settings [95].

### 4.6. Gene Family Members and Promoter Sequence Analysis

The homologs of the five floral induction and meristem identity genes *FT*, *SOC1*, *LFY*, *AP1*, and *FUL* in the *A. thaliana* genome were identified. Whole gene sequences, including coding and promoter regions of these homologs, were downloaded from Phytozome v13 [96] and the NCBI database (https://www.ncbi.nlm.nih.gov/; accessed on 19 December 2022). The 1 Kb upstream promoter region of the cotton homologs of *Arabidopsis* genes was analyzed using PlantPAN v. 3.0 [97], with default parameters for transcription-factor-binding sites. We also studied the whole gene and the promoter sequences of the cotton gene homologs of *Arabidopsis* genes (mentioned above) and the genes with expression-trait-associated markers (identified through the association analysis) for the presence of pre-miRNAs and miRNA target sites. Additionally, we analyzed these whole gene sequences for the presence of CpG islands (CGIs). To predict CGIs, the following criteria were used on CpGPlot v. 6.6.0 (www.ebi.ac.uk/Tools/emboss/cpgplot; accessed on 21 December 2022): observed/expected ratio of ≥0.60, percentage C/G nucleotide ≥ 50.0%, and length ≥ 200 bp. The genes with expression-trait-associated markers were functionally annotated by blasting them against the NCBI nr protein database, and the presence of miRNA coding genes was predicted by blasting them against the miRBase database release 22.1 [98]. Subsequently, the genes were classified into different classes based on their functions manually. The expression profiles of the five floral induction and meristem identity genes and the genes with expression trait-associated markers were studied by blasting the whole gene sequences against the cDNA/EST sequences at NCBI. A hierarchical clustering analysis of gene expression data was performed and visualized using InstantClue v. 0.10.10.dev-snap [99].

## 5. Conclusions

In this study, we demonstrated genome-wide association analysis on 44 Upland cotton mini-core collection genotypes to identify eQTLs for five floral induction and meristem identity genes that were earlier shown to integrate the environmental and internal cues to guide meristems to either continue vegetative growth or culminate it by flower production. This analysis was performed to develop molecular markers and identify genotypes to breed for optimal plant architecture, improved lint yield, and reduced regrowth after defoliation. The identification of coincident and consistent eQTLs indicated the application of a modest-size population in identifying QTLs for e-traits and genotypes with desirable alleles for genetic crossing and gene pyramiding. Further, in this study, we were able to predict the molecular functions of eQTLs and identify the molecular cause of epistatic interactions among eQTLs and genes of interest. This information may serve as a resource for the breeding community to develop high-yielding cotton genotypes with optimal architecture suited to different production environments. Developing genotypes tailored to specific production environments is possible using the quantitative genetic approach adopted here. On the contrary, silencing or ectopic expression of the major floral induction and meristem identity genes attempted earlier resulted in extreme phenotypes with adverse impacts on yield. This moderate approach with the ability to finetune the phenotype was earlier adopted by Rodríguez-Leal et al. [100] by engineering quantitative trait variation in tomato by editing promoters of specific genes. The current approach relied on pyramiding the effects of the eQTLs on the expression levels of five floral induction and meristem identity genes to develop cotton genotypes with variable degrees of determinacy with improved yield and reduced regrowth. This approach acts on cotton architecture to maximize yield via channeling resources towards developing bolls by tweaking cutout timing. It would reduce the input requirements, specifically the amount of plant growth regulators used to alter the architecture or terminate the crop. In sum, the architecturally altered cotton will maximally utilize its potential towards actual yield, avoid regrowth, and, as stated, reduce input; hence, it is expected to improve returns. However, the major expected challenges associated with such cotton types would be a change in the harvesting regimen due to changes in the plant architecture, such as reduced height due to the shortening of internode length and floral clustering. Also, changes in flowering time and duration are expected, requiring adjustments in crop management practices. Further, the change in climatic conditions, specifically heat and drought, may impact the performance of such cotton types though their impact on the timing, length, and severity of the stress.

## Figures and Tables

**Figure 1 ijms-24-14174-f001:**
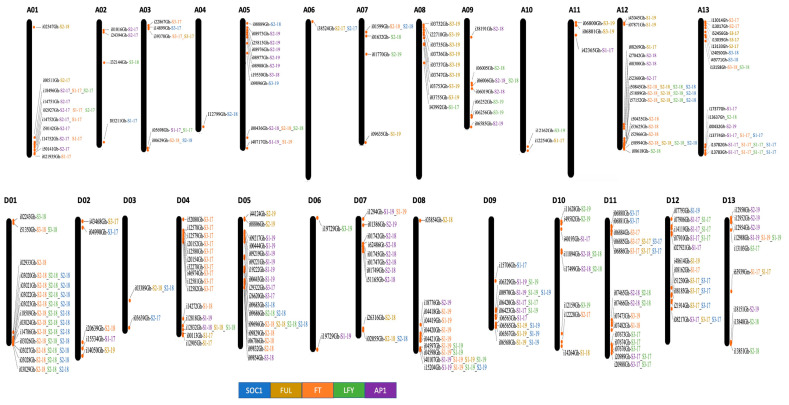
A map of cotton chromosomes (A and D subgenomes) showing locations of expression QTLs (eQTLs). S1, S2, and S3 represent three developmental stages at which the expression level of the cotton *FT*, *SOC1*, *LFY*, *FUL*, and *AP1* genes was studied. Different colors represent eQTLs detected for the genes of interest, and detection years are marked as a suffix (17, 18, and 19) to the marker name. The A subgenome chromosomes are designated A01–A13, and the D subgenome chromosomes D01–D013.

**Figure 2 ijms-24-14174-f002:**
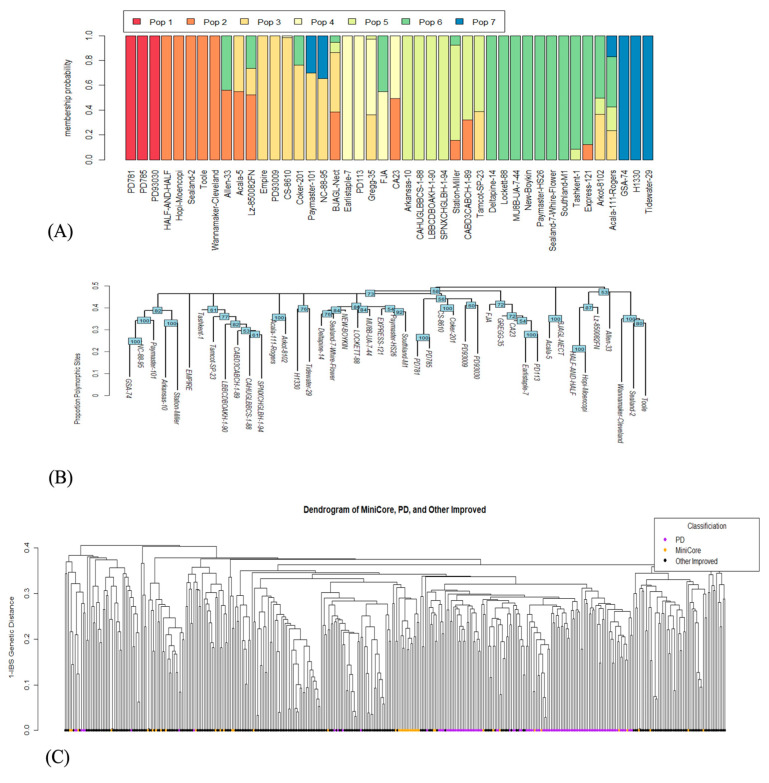
Figure showing interrelationships of the 44 Upland cotton genotypes of the mini-core collection and the genotypes of a larger collection of Upland cotton genotypes. Population structure analysis of the 44 Upland cotton genotypes, (**A**). Dendrogram of 44 Upland cotton genotypes developed based on the proportion of polymorphic sites, (**B**). Dendrogram of the Upland cotton genotypes ever cultivated in Southeastern US, (**C**). Nodes with orange circles represent 44 Upland cotton genotypes of the mini-core collection, purple circles represent Pee Dee lines, and black nodes represent other improved germplasm under cultivation.

**Table 1 ijms-24-14174-t001:** List of top-scoring genotypes in the gene expression study selected for genetic crossing.

ID	Genotype	Points 2017	Points 2018	Points 2019	Total Points
8	CABD3CABCH-1-89	14	7	13	34 (76%)
5	ARKOT 8102	9	8	15	32 (71%)
20	HOPI MOENCOPI	11	10	11	32 (71%)
37	SPNXCHGLBH-1-94	12	8	8	28 (62%)
9	CAHUGLBBCS-1-88	9	6	8	23 (51%)
39	TAMCOT SP-23	6	9	8	23 (51%)
10	COKER 201	9	9	4	22 (49%)
17	GSA 74	4	8	7	19 (42%)

**Table 2 ijms-24-14174-t002:** List of the candidate genes identified from the expression QTL analysis of *FT*, *SOC1*, *LFY*, *FUL*, and *AP1* cotton genes using Upland cotton mini-core collection.

Sr. No.	Molecular Markers	Gene ID	Sub-Genome	Gene Name	Position on Genome (Mb)	Annotation
1	i02927Gh	Gohir.A01G208700	A01	*AP1*, *FT*, *LFY*	117.19592	Trihelix transcription factor PTL
2	i43992Gh	Gohir.A08G034500	A08	*FT*	4.35425	MYB3-like transcription factor
3	i13158Gh	Gohir.A13G050400	A13	*FT*, *LFY*	7.06423	GATA transcription factor 28-like
4	i09222Gh; i00443Gh	Gohir.D05G103700	D05	*AP1*	8.73032	GATA transcription factor 11-like
5	i08185Gh	Gohir.D12G153600	D12	*FUL*, *SOC1*	48.44348	SQUAMOSA promoter binding-like transcription factor
6	i13848Gh; i13851Gh	Gohir.D13G236200	D13	*LFY*	64.67230	Homeobox-leucine zipper protein REVOLUTA-like

## Data Availability

The data presented in this paper are available on request from the corresponding author.

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
