# Peer review of "Alterations in Growth Habit to Channel End-of-Season Perennial Reserves towards Increased Yield and Reduced Regrowth after Defoliation in Upland Cotton (Gossypium hirsutum L.)"

_ijms, 2023, doi:10.3390/ijms241814174_

Round 1

Reviewer 1 Report

Review Report: ijms-2572367

The manuscript, titled "Alterations in Growth Habit to Channel End-of-season Perennial Reserves Towards Increased Yield and Reduced Regrowth After Defoliation in Upland Cotton (Gossypium hirsutum L.)” presents a study examining how altering upland cotton plants' growth habits affects end-of-season perennial reserves, enhances yield, and reduces regrowth after defoliation. The authors delve into crucial facets of cotton cultivation and its reaction to defoliation, providing insightful perspectives on optimizing cotton yield and curtailing regrowth. The study is well-executed and makes substantial contributions to the realm of cotton agriculture. A comprehensive review of the manuscript follows.

The study tackles a crucial concern within cotton agriculture: optimizing yield and reducing regrowth after defoliation. The outcomes hold pragmatic significance for cotton farmers aiming to refine their approaches to crop management.

Introduction:

Although the introduction establishes a broad context for the study, enhancing it could involve incorporating a more comprehensive overview of the difficulties arising from regrowth following defoliation in cotton cultivation. Furthermore, the introduction would gain from a succinct elucidation of the physiological foundation of end-of-season reserves and their correlation with cotton yield.

Objective:

The objectives of the manuscript are clearly articulated, centering on modifying growth habits to increase cotton yield and reduce post-defoliation regrowth.

The results section succinctly presents significant findings; nonetheless, certain figures lack adequate quality and necessitate improvement to enhance their clarity.

Within the discussion section, exploring the potential physiological mechanisms that underlie the observed modifications in growth habits would be highly beneficial. This exploration might encompass an examination of hormonal regulation, allocation of resources, and genetic elements that play a role in the documented transformations.

Similarly, in the conclusion section, there is a need to add practical implications to augment the manuscript's real-world applicability and contemplate a discussion on the viability of implementing growth habit adjustments on a broader scale. Additionally, acknowledge possible obstacles or constraints that cotton producers could confront when endeavoring to embrace these strategies.

Minor edits.

a [84]. were synthesized and used in qRT-PCR

following Hulse-Kemp et al. (2015).

Figure 1 and Fig 2C is blur, there is a need to change the figure to high-quality resolution.

Line 363-365 rephrase the sentence for the intended meaning

Carefully proofread the manuscript for grammatical and typographical errors.

Ensure consistent formatting of references according to the journal's guidelines.

Minor edits requires 

Reviewer 2 Report

The manuscript with the title “Alterations in Growth Habit to Channel End-of-season Perennial Reserves Towards Increased Yield and Reduced Regrowth After Defoliation in Upland Cotton (Gossypium hirsutum L.)” explored genome-wide association involved in the cotton growth habit. Authors studied 44 cotton genotypes from the most economically important species of cotton.

The manuscript is written in good English and overall has a logical flow.
However, at the end of the introduction, I advise authors to introduce a paragraph where they clearly express the aim and objectives of their research. Given the length of their manuscript, there should be one aim and several objectives (as steps proposed for reaching the aim). This paragraph will help readers to anticipate what they are about to read. The objectives should be also reflected in structure and order of results subsections and conclusion section. Each objective shall have a corresponding conclusion at the end, preferably answered in the same order.  

Introduction provides adequate background while material and methods are sufficiently detailed.
Results section with 10 subchapters, presents the findings. Figures have too low resolution, this should be addressed.

Conclusions – usually they mirror the objectives of the study. We do not know if the authors reached/solved their objectives because these were not clearly expressed at the start.

References
Out of 97 titles cited, 1 paper is from 2023, 4 are from 2022, 1 from 2021 and the rest all older. References are on the topic, but a better connection with the current/recent body of literature might be needed by citing a few more recent reputable papers on the subject.

Best regards.

Round 2

Reviewer 1 Report

The authors respond well to all the suggested improvements in the manuscript. The revised version has been found suitable for publication in the IJMS journal.